# FLOW-BASED RECURRENT BELIEF STATE LEARNING FOR POMDPs

## ABSTRACT

Partially Observable Markov Decision Process (POMDP) provides a principled and generic framework to model real world sequential decision making processes but yet remains unsolved, especially for high dimensional continuous space and unknown models. The main challenge lies in how to accurately obtain the belief state, which is the probability distribution over the unobservable environment states given historical information. Accurately calculating this belief state is a precondition for obtaining an optimal policy of POMDPs. Recent advances in deep learning techniques show great potential to learn good belief states, but they assume the belief states follow certain types of simple distributions such as diagonal Gaussian, which imposes strong restrictions to precisely capture the real belief states. In this paper, we introduce the **Fl**O**w-based **R**ecurrent **BE**lief **S**tate model (FORBES), which incorporates normalizing flows into the variational inference to learn general continuous belief states for POMDPs. Furthermore, we show that the learned belief states can be plugged into downstream RL algorithms to improve performance. In experiments, we show that our methods successfully capture the complex belief states that enable multi-modal predictions as well as high quality reconstructions, and results on challenging visual-motor control tasks show that our method achieves superior performance and sample efficiency.

## 1 INTRODUCTION

Partially Observable Markov Decision Process (POMDP) (Åström, 1965) provides a principled and generic framework to model real world sequential decision making processes. Unlike Markov Decision Process (MDP), the observations of a POMDP are generally non-Markovian. Therefore, to make optimal decisions, the agent needs to consider all historical information, which is usually intractable. One effective solution is to obtain the belief state. The belief state is defined as the probability distribution of the unobservable environment state conditioned on the past observations and actions (Kaelbling et al., 1998). Such belief state accurately summarizes the history. Traditional methods of calculating belief states (Smallwood & Sondik, 1973; Sondik, 1971; Kaelbling et al., 1998) assume finite discrete space with a known model. In many real world problems, however, the underlying model remains unknown, and the state space is large and even continuous.

With the recent advances of deep learning technologies, a branch of works have been proposed to learn the belief states of POMDPs with unknown model and continuous state space (Krishnan et al., 2015; Gregor et al., 2019; Lee et al., 2020; Hafner et al., 2019b;a; 2021). These works solve the belief state learning problem by sequentially maximizing the observation probability at each timestep using the variational inference and achieve the state-of-the-art performance on many visual-motor control tasks (Hafner et al., 2019a; Zhu et al., 2020; Okada et al., 2020; Ma et al., 2020a). However, they still cannot capture general belief states due to the intractability of complex distributions in high-dimensional continuous space and instead approximate the belief states with diagonal Gaussians. This approximation imposes strong restrictions and is problematic. As shown in Figure 1, the blue area denotes the unobservable state space of the POMDP. Given the past information $\tau$, the agent maintains a prior distribution of the state $s$, denoted as $p(s|\tau)$ (the distribution in white). Each colored distribution corresponds to the belief state after receiving a different new observation $o$, denoted as the posterior distribution $q(s|\tau, o)$. Consider an example of the true beliefs as shown in Figure 1(b), with their Gaussian approximations shown in Figure 1(a). The approximation error of Gaussian distributions will easily result in problems of intersecting belief which leads to a mixed-up

state (e.g., the white triangle), and empty belief, which leads to a meaningless state (e.g., the grey triangle). This also explains the poor reconstruction problems in interactive environments observed by Okada & Taniguchi (2021). Furthermore, as mentioned in Hafner et al. (2021), the Gaussian approximation of belief states also makes it difficult to predict multi-modal future behaviours. Therefore, it is preferable to relax the Gaussian assumptions and use a more flexible family of distributions to learn accurate belief states as shown in Figure 1(b).

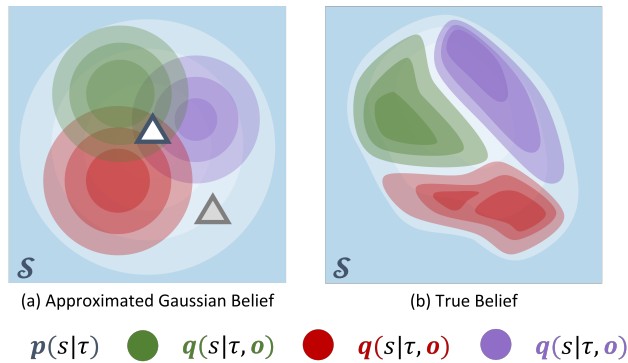

(a) Approximated Gaussian Belief      (b) True Belief

$p(s|\tau)$    $q(s|\tau, o)$    $q(s|\tau, o)$    $q(s|\tau, o)$

Figure 1: Difference between (a) spherical Gaussian belief states and (b) true belief states (Better viewed in color).

In this paper, we propose a new method called **Fl**O**w**-based **R**ecurrent **BE**lief **S**tate model (FORBES) that is able to learn general continuous belief states for POMDPs. FORBES incorporates Normalizing Flows (Tabak & Turner, 2013; Rezende & Mohamed, 2015; Dinh et al., 2017) into the variational inference step to construct flexible belief states. In experiments, we show that FORBES allows the agent to maintain flexible belief states, which result in multi-modal and precise predictions as well as higher quality reconstructions. We also demonstrate the results combining FORBES with downstream RL algorithms on challenging visual-motor control tasks (DeepMind Control Suite, Tassa et al. (2018)). The results show the efficacy of FORBES in terms of improving both performance and sample efficiency.

Our contributions can be summarized as follows:

- We propose FORBES, the first flow-based belief state learning algorithm that is capable of learning general continuous belief states for POMDPs.
- We propose a POMDP RL framework based on FORBES for visual-motor control tasks, which uses the learned belief states from FORBES as the inputs to the downstream RL algorithms.
- Empirically, we show that FORBES allows the agent to learn flexible belief states that enable multi-modal predictions as well as high quality reconstructions and help improve both performance and sample efficiency for challenging visual-motor control tasks.

## 2 PRELIMINARIES

### 2.1 PARTIALLY OBSERVABLE MARKOV DECISION PROCESS

Formally, a Partially Observable Markov Decision Process (POMDP) is a 7-tuple $(\mathcal{S}, \mathcal{A}, T, R, \Omega, O, \gamma)$, where $\mathcal{S}$ is a set of states, $\mathcal{A}$ is a set of actions, $T$ is a set of conditional transition probabilities between states, $R$ is the reward function, $\Omega$ is a set of observations, $O$ is a set of conditional observation probabilities, and $\gamma$ is the discount factor.

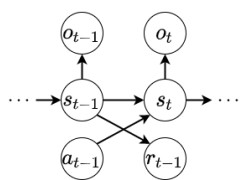

Figure 2: The PGM of POMDP

At each timestep $t - 1$, the state of the environment is $s_{t-1} \in \mathcal{S}$. The agent takes an action $a_{t-1} \in \mathcal{A}$, which causes the environment to transit to state $s_t$ with probability $T(s_t \mid s_{t-1}, a_{t-1})$. The agent then receives an observation $o_t \in \Omega$ which depends on the new state of the environment $s_t$ with probability $O(o_t \mid s_t)$. Finally, the agent receives a reward $r_{t-1}$ equal to $R(s_{t-1})$. The agent's goal is to maximize the the expected

sum of discounted rewards $\mathbb{E}\left[\sum_{t=0}^{\infty} \gamma^t r_t\right]$. Such a POMDP model can also be described as a probabilistic graphical model (PGM) as shown in Figure 2. After having taken action $a_{t-1}$ and observing $o_t$, an agent needs to update its belief state, which is defined as the probability distribution of the environment state conditioned on all historical information:

$$b(s_t) = p(s_t \mid \tau_t, o_t) \tag{1}$$

where $\tau_t = \{o_1, a_1, \ldots, o_{t-1}, a_{t-1}\}$.

## 2.2 NORMALIZING FLOW

Instead of using the Gaussian family to approximate the prior and posterior belief distributions, we believe it is more desirable to use a family of distributions that is highly flexible and preferably flexible enough to describe all possible true belief states. Therefore, we use Normalizing Flows (Tabak & Turner, 2013; Rezende & Mohamed, 2015) to parameterize those distributions.

Rather than directly parameterizing statistics of the distribution itself, Normalizing Flows model the transformations, or the "flow" progress, needed to derive such a distribution. More specifically, it describes a sequence of invertible mappings that gradually transform a relatively simple probability density to a more flexible and complex one.

Let $f_\theta : \mathbb{R}^D \to \mathbb{R}^D$ to be an invertible and differentiable mapping in state space parameterized by $\theta$. Given a random variable $\mathbf{x} \in \mathbb{R}^D$ with probability distribution $p(\mathbf{x})$, we can derive the probability of the transformed random variable $\mathbf{z} = f_\theta(\mathbf{x})$ by applying the change of variable formula:

$$p(\mathbf{z}) = p(\mathbf{x}) \left| \det \frac{\partial f_\theta^{-1}}{\partial \mathbf{z}} \right| \tag{2}$$

$$\log p(\mathbf{z}) = \log p(\mathbf{x}) - \log \left| \det \frac{\partial f_\theta}{\partial \mathbf{z}} \right| \tag{3}$$

To construct a highly flexible family of distributions, we can propagate the random variable at beginning $\mathbf{z}_0$ through a sequence of $K$ mappings and get $\mathbf{z}_K = f_{\theta_K} \circ f_{\theta_{K-1}} \circ \cdots \circ f_{\theta_1}(\mathbf{z}_0)$ with the probability

$$\log p_K(\mathbf{z}_K) = \log p(\mathbf{z}_0) - \sum_{k=1}^{K} \log \left| \det \frac{\partial f_{\theta_k}}{\partial \mathbf{z}_{k-1}} \right| \tag{4}$$

Given a relatively simple distribution of $\mathbf{z}_0$, say, Gaussian distribution, by iteratively applying the transformations, the flow is capable of representing a highly complex distribution with the probability that remains tractable. The parameters $\theta_1, \ldots, \theta_K$ determine the transformations of the flow.

An effective transformation that is widely accepted is affine coupling layer (Dinh et al., 2017; Kingma & Dhariwal, 2018; Kingma et al., 2017). Given the input $\mathbf{x} \in \mathbb{R}^D$, let $s$ and $t$ stand for scale and translation functions which are usually parameterized by neural networks, where $s, t : \mathbb{R}^k \to \mathbb{R}^{D-k}, k < D$. The output, $\mathbf{y}$, can be viewed as a concatenation of its first $k$ dimensions $\mathbf{y}_{1:k}$ and the remaining part $\mathbf{y}_{k+1:D}$:

$$\mathbf{y}_{1:k} = \mathbf{x}_{1:k}, \quad \mathbf{y}_{k+1:D} = \mathbf{x}_{k+1:D} \odot \exp(s(\mathbf{x}_{1:k})) + t(\mathbf{x}_{1:k}) \tag{5}$$

where $\odot$ denotes the element-wise product (see details about affine coupling layer in Appendix A.1).

# 3 FLOW-BASED RECURRENT BELIEF STATE LEARNING

## 3.1 FLOW-BASED RECURRENT BELIEF STATE MODEL

We propose the **Fl**Ow-based **R**ecurrent **BE**lief **S**tate model (FORBES) which learns general continuous belief states via normalizing flows under the variational inference framework. Specifically, the FORBES model consists of components needed to construct the PGM of POMDP as shown in Figure 2:

$$
\begin{aligned}
\text{State transition model}: \quad & p(s_t|s_{t-1}, a_{t-1}) \\
\text{Observation model}: \quad & p(o_t|s_t) \\
\text{Reward model}: \quad & p(r_t|s_t)
\end{aligned}
\tag{6}
$$

In addition, we have a belief inference model $q(s_t|\tau_t, o_t)$ to approximate the true posterior distribution $p(s_t|\tau_t, o_t)$, where $\tau_t = \{o_1, a_1, \ldots, o_{t-1}, a_{t-1}\}$ is the past information. The above components of FORBES can be optimized jointly by maximizing the Evidence Lower BOund (ELBO) (Jordan et al., 1999) or more generally the variational information bottleneck (Tishby et al., 2000; Alemi et al., 2016):

$$
\begin{aligned}
&\log p(o_{1:T}, r_{1:T}|a_{1:T}) \\
&\geq \mathbb{E}_{q(s_{1:T}|o_{1:T}, a_{1:T-1})} \left[ \sum_{t=1}^{T} (\ln p(o_t|s_t) + \ln p(r_t|s_t) - D_{\mathrm{KL}}(q(s_t|s_{t-1}, a_{t-1}, o_t) \| p(s_t|s_{t-1}, a_{t-1}))) \right] \\
&= \sum_{t=1}^{T} \Big[ \mathbb{E}_{q(s_t|s_{t-1}, a_{t-1}, o_t)}(\ln p(o_t|s_t) + \ln p(r_t|s_t)) - \\
&\qquad\qquad \mathbb{E}_{q(s_{t-1}|s_{t-2}, a_{t-2}, o_{t-1})}(D_{\mathrm{KL}}(q(s_t|s_{t-1}, a_{t-1}, o_t) \| p(s_t|s_{t-1}, a_{t-1}))) \Big] \doteq \mathcal{J}_{\mathrm{Model}}
\end{aligned}
\tag{7}
$$

here we use the factorization of $q(s_{1:T}|o_{1:T}, a_{1:T-1}) = \prod_t q(s_t|s_{t-1}, a_{t-1}, o_t)$.

Detailed derivations can be found in Appendix.A.9. In practice, the state transition model, observation model, reward model, and belief inference model can be represented by stochastic deep neural networks parameterized by $\psi$:

$$
p_\psi(s_t|s_{t-1}, a_{t-1}), \quad p_\psi(o_t|s_t), \quad p_\psi(r_t|s_t), \quad q_\psi(s_t|\tau_t, o_t)
\tag{8}
$$

where their outputs usually follow simple distributions such as diagonal Gaussians. The parameterized belief inference model $q_\psi(s_t|\tau_t, o_t)$ acts as an encoder that encodes the historical information using a combination of convolutional neural networks and recurrent neural networks. Note that $q$ below is obtained through the factorization of $q(s_{1:T}|o_{1:T}, a_{1:T-1}) = \prod_t q(s_t|s_{t-1}, a_{t-1}, o_t)$.

In FORBES we provide special treatments for the belief inference model and the state transition model to represent more complex and flexible posterior and prior distributions. As shown in Figure 3(a), the input images $o_{1:t}$ and actions $a_{1:t-1}$ are encoded with $q_\psi(s_t|\tau_t, o_t)$ (the blue and the red path). Then our final inferred belief is obtained by propagating $q_\psi(s_t|\tau_t, o_t)$ through a set of normalizing flow mappings denoted $f_{\theta_K} \circ \cdots \circ f_{\theta_1}$ to get a representative posterior distribution $q_{\psi,\theta}(s_t|\tau_t, o_t)$. For convenience, we denote $q_0 = q_\psi$ and $q_K = q_{\psi,\theta}$. On the other hand, $o_{1:t-1}$ and $a_{1:t-2}$ are encoded with $q_\psi(s_{t-1}|\tau_{t-1}, o_{t-1})$ (the blue path), then the state transition model is used to obtain the prior guess of the state $p_\psi(s_t \mid \tau_t) = \mathbb{E}_{q_\psi(s_{t-1}|\tau_{t-1}, o_{t-1})}[p_\psi(s_t \mid s_{t-1}, a_{t-1})]$ (the green path). Then our final prior is obtained by propagating $p_\psi(s_t|\tau_t)$ through another set of normalizing flow mappings denoted $f_{\omega_K} \circ \cdots \circ f_{\omega_1}$ to get a representative prior distribution $p_{\psi,\omega}(s_t|\tau_t)$. For convenience, we denote $p_0 = p_\psi$ and $p_K = p_{\psi,\omega}$. Then as shown in Figure 3(b), we can sample the initial state $s_t$ (the yellow and purple triangles) from the belief states $q_K(s_t \mid \tau_t, o_t)$. For each sampled initial state, we can use the state transition model to predict the future states $\hat{s}_{t+h}$ given the future actions $a_{t:t+h-1}$, and then use the observation model to reconstruct the observations $\hat{o}_{t+h}$, where $h$ is the prediction horizon.

With the above settings, we can substitute the density probability inside the KL-divergence term in Equation 7 with Normalizing Flow:

$$
\begin{aligned}
&D_{\mathrm{KL}}(q_K(s_t|\tau_t, o_t) \| p_K(s_t|s_{t-1}, a_{t-1})) = \mathbb{E}_{q_K(s_{1:t}|\tau_t, o_t)}[\log q_K(s_t|\tau_t, o_t) - \log p_K(s_t|\tau_t)] \\
&= \mathbb{E}_{q_K(s_{1:t}|\tau_t, o_t)} \left[ \log q_0(s_t|\tau_t, o_t) - \sum_{k=1}^{K} \log \left| \det \frac{\partial f_{\theta_k}}{\partial s_{t,k-1}} \right| - \log p_0(s_t|\tau_t) + \log \left| \det \frac{\partial f_{\omega_k}}{\partial s_{t,k-1}} \right| \right]
\end{aligned}
\tag{9}
$$

where $p_K(s_t \mid s_{t-1}, a_{t-1}) = p_K(s_t \mid \tau_t)$ given the sampled $s_{t-1}$ from $q_K(s_{1:t}|\tau_t, o_t)$. $s_{t,k}$ is the state variable $s_t$ transformed by $k$ layers of normalizing flows, and $s_{t,0} = s_t$.

To further demonstrate the properties of FORBES, we provide the following theorems.

**Theorem 1** *The approximation error of the log-likelihood when maximizing the $\mathcal{J}_{\mathrm{Model}}$ (the derived ELBO) defined in Equation 7 is:*

$$
\log p(o_{1:T}, r_{1:T}|a_{1:T}) - \mathcal{J}_{\mathrm{Model}} = \mathbb{E}_{q_K(s_{1:T}|\tau_T, o_T)} \left[ \sum_{t=1}^{T} D_{\mathrm{KL}}(q(s_t|\tau_t, o_t) \| p(s_t \mid \tau_t, o_t)) \right]
\tag{10}
$$

*where $p(s_t \mid \tau_t, o_t)$ denotes the true belief states.*

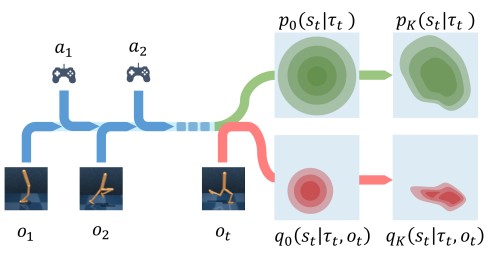 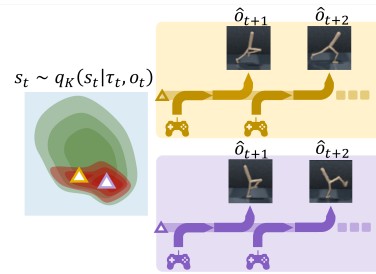

(a) Belief state inference    (b) Predictions beginning from different samples

Figure 3: The algorithm framework of FORBES. Figure 3a shows how to calculate prior and posterior belief distribution given previous information. The blue arrows bring in historical observations and actions, and the green path shows the evolution of prior belief distribution. The red path takes an additional $o_t$ and shows the evolution of posterior belief distribution. Figure 3b shows prediction of future trajectories given the future actions.

Detailed proofs can be found in Appendix.A.9. Theorem 1 suggests that, when the learning algorithm maximizes the $\mathcal{J}_{\text{Model}}$ (the derived ELBO), then the $D_{\text{KL}}$ terms in the right-hand side are minimized, which indicate the KL-divergence between the learned belief states $q(s_t|\tau_t, o_t)$ and the true belief states $p(s_t \mid \tau_t, o_t)$. Clearly, if $p(s_t \mid \tau_t, o_t)$ is a complex distribution and $q(s_t|\tau_t, o_t)$ is chosen from a restricted distribution class such as diagonal Gaussian, then when the algorithm maximizes the $\mathcal{J}_{\text{Model}}$ (the derived ELBO), there will still be a potentially large KL-divergence between the learned and the true belief states. Therefore, naturally there raises the problem that is normalizing flow a universal distributional approximator that is capable of accurately representing arbitrarily complex belief states, so the KL-divergence terms in the right-hand side of Equation (10) can be minimized to approach zero? The answer is yes for a wide range of normalizing flows. To be specific, Teshima et al. (2020) provides theoretical results for the family of the flow used in FORBES.

In fact, there always exists a diffeomorphism that can turn one well-behaved distribution to another. Besides the aforementioned affine coupling flow, many works show the distributional universality of other flows (Kong & Chaudhuri, 2020; Huang et al., 2018). Ideally, the universal approximation property of the flow model $q_K(s_t \mid \tau_t, o_t)$ allows us to approximate the true posterior $p(s_t \mid \tau_t, o_t)$ with arbitrary accuracy. Thus, compared to previous methods, FORBES helps close the gap between the log-likelihood and the ELBO to obtain a more accurate belief state. Though we usually cannot achieve the ideal zero KL-divergence in practice, our method can get a smaller approximation error, equally a higher ELBO than previous works. We verify this statement in section 4.1.

## 3.2    POMDP RL FRAMEWORK BASED ON FORBES

To show the advantage of the belief states inferred by the FORBES model compared to the existing belief inference method in visual-motor control tasks, we design a flow-based belief reinforcement learning algorithm for learning the optimal policy in POMDPs. The algorithm follows an actor-critic framework. The critic estimates the accumulated future rewards, and the actor chooses actions to maximize the estimated cumulated rewards. Both the actor and critic operate on top of the samples of learned belief states, and thus benefit from the accurate representations learned by the FORBES model. Note that this is an approximation of the true value on belief, which avoids the intractable integration through observation model.

The critic $v_\xi(s_\tau)$ aims to predict the discounted sum of future rewards that the actor can achieve given an initial state $s_t$, known as the state value $\mathbb{E}\left(\sum_{\tau=t}^{t+\infty} \gamma^{\tau-t} r_\tau\right)$, where $\xi$ denote the parameters of the critic network and $H$ is the prediction horizon. We leverage temporal-difference to learn this value, where the critic is trained towards a value target that is constructed from the intermediate reward and the critic output for the next step's state. In order to trade-off the bias and the variance of the state value estimation, we use the more general TD($\lambda$) target (Sutton & Barto, 2018), which is a weighted average of n-step returns for different horizons and is defined as follows:

$$V_\tau^\lambda \doteq \hat{r}_\tau + \hat{\gamma}_\tau \begin{cases} (1-\lambda)v_\xi(s_{\tau+1}) + \lambda V_{\tau+1}^\lambda & \text{if} \quad \tau < t + H, \\ v_\xi(s_{t+H}) & \text{if} \quad \tau = t + H. \end{cases} \tag{11}$$

To better utilize the flexibility belief states from FORBES, we run the sampling method multiple times to capture the diverse predictions. Specifically, we sample $N$ states from the belief state given by FORBES and then rollout trajectories of future states and rewards using the state transition model and the reward model. Finally, we train the critic to regress the TD($\lambda$) target return using a mean squared error loss:

$$\mathcal{J}_{\text{Critic}}(\xi) = \mathbb{E}_{s_{i,0} \sim q_K, a_\tau \sim q_\phi, s_{i,\tau} \sim p_\psi} \left[ \sum_{i=1}^N \sum_{\tau=t}^{t+H} \frac{1}{2} \big( v_\xi(s_{i,\tau}) - \text{sg}(V_{i,\tau}^\lambda) \big)^2 \right]. \tag{12}$$

where $sg(\cdot)$ is the stop gradient operation. The actor $a_\tau \sim q_\phi(a_\tau \mid s_\tau)$ aims to output actions that maximize the prediction of long-term future rewards made by the critic and is trained directly by backpropagating the value gradients through the sequence of sampled states and actions, i.e., maximize:

$$\mathcal{J}_{\text{Actor}}(\phi) = \mathbb{E}_{s_{i,0} \sim q_K, a_\tau \sim q_\phi, s_{i,\tau} \sim p_\psi} \left( \sum_{i=1}^N \sum_{\tau=t}^{t+H} V_{i,\tau}^\lambda \right) \tag{13}$$

We jointly optimize the model loss $\mathcal{J}_{Model}$ with respect to the model parameters $\psi$, $\theta$ and $\omega$, the critic loss $\mathcal{J}_{Critic}$ with respect to the critic parameters $\xi$ and the actor $\mathcal{J}_{Actor}$ loss with respect to the actor parameters $\phi$ using the Adam optimizer with different learning rates:

$$\min_{\psi, \xi, \phi, \theta, \omega} \mathcal{J}_{\text{FORBES}} = \alpha_0 \mathcal{J}_{\text{Critc}}(\xi) - \alpha_1 \mathcal{J}_{\text{Actor}}(\phi) - \alpha_2 \mathcal{J}_{\text{Model}}(\psi, \theta, \omega) \tag{14}$$

where $\alpha_0$, $\alpha_1$, $\alpha_2$ are coefficients for different components, and we summarize the whole framework of optimizing in Algorithm 1.

---

**Algorithm 1** FORBES Algorithm

---

**Input: buffer $\mathcal{B}$, imagination horizon $H$, interacting step $T$, batch size $B$, batch length $L$, number of trajetories $N$.**

Draw $B$ data sequences $\{(o_t, a_t, r_t)\}_{t=k}^{k+L}$ from $\mathcal{B}$

Infer belief state $q_K(s_t|\tau_t, o_t)$.

**for** $i = 1, \ldots, N$ **do**

Rollout imaginary trajectories $\{(s_{i,\tau}, a_{i,\tau})\}_{\tau=t}^{t+H}$ with belief transition model.

**end for**

For each $s_{i,\tau}$, predict rewards $p_\psi(r_{i,\tau}|s_{i,\tau})$ and values $v_\phi(s_{i,\tau})$ ▷ *Calculate imaginary returns*

Update $\theta, \omega, \xi, \phi, \psi$ using Equation (7), (9), (12), (13) and (14) ▷ *Optimize the model, critic and actor*

---

## 4 EXPERIMENTS

Our experiments evaluate FORBES on two image-based tasks. We first demonstrate the belief learning capacity on a digit writing task in Section 4.1, and show that FORBES captures beliefs that allow for multi-modal yet precise long-term predictions as well as higher ELBO. For large-scale experiments, we test the proposed POMDP RL framework based on FORBES in Section 4.2. The results of multiple challenging visual-motor control tasks from DeepMind Control Suite (Tassa et al., 2018) show that FORBES outperforms baselines in terms of performance and sample efficiency. In Section 4.3, we further provide ablation studies of the multiple imagined trajectories technique used in our method.

### 4.1 DIGIT WRITING TASKS

In this experiment, we validate the capacity of FORBES by modelling the partially observable sequence with visual inputs. We adopt the MNIST Sequence Dataset (D. De Jong, 2016) that consists of sequences of handwriting MNIST digit stokes. This problem can be viewed as a special case

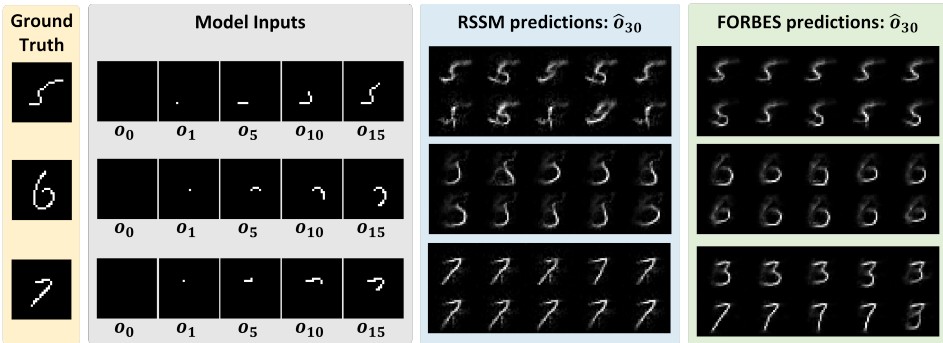

Figure 4: Predictions on sequential MNIST of two models.

of POMDP, whose action space is Ø and rewards remain 0. Such a problem setting separates the belief learning and policy optimizing problem and allows us to concentrate on the former one in this section. We convert the digit stroke to a sequence of images of size $28 \times 28$ to simulate the writing process. At time step $t$, the agent can observe $o_t$ that has already written $t$ pixels, and we train the agent maximizing $\mathcal{J}_{\text{Model}}$ in Equation 7 except for the reward reconstruction term.

As shown in Figure 4, we randomly select three digits as examples (see Appendix A.10 for more results) and show the inputs as well as the prediction outputs of our model and the RSSM (Hafner et al., 2019b) baseline, which is the previous state-of-the-art method for learning continuous belief states of POMDPs. The leftmost column is the ground truth of the fully written digits. During the testing, we feed the initial 15 frames $\{o_1, o_2, \cdots, o_{15}\}$ to the model, and the columns in grey exhibit a part of the inputs. Then we sample several states from the inferred belief state and rollout via the learned state transition model (Equation (6)) for 15 steps and show the reconstruction results of the predictions. As shown in the blue and green columns on the right of Figure 4, though RSSM can also predict the future strokes in general, the reconstructions are relatively blurred and mix different digits up. It also fails to give diverse predictions. However, FORBES can make precise yet diverse predictions. Each

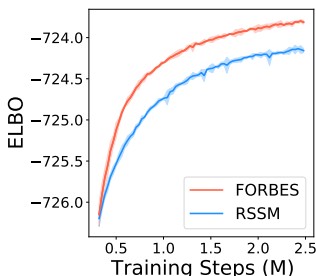

Figure 5: ELBO on digit writing

prediction is clear and distinct from other digits. Given the beginning of the digit 7, FORBES successfully predicts both 7 and 3 since they have a similar beginning. The results can be partially explained via the mixed-up belief and the empty belief as shown in Figure 1, which support the claim that FORBES can better capture the complex belief states.

We also provide the quantitative results in Figure 5, which is the ELBO on test digits sequence set that is never seen during training. The results show that FORBES can achieve a tighter ELBO, which verifies the theoretical results in 3.1. Details of the network can be found in Appendix A.2.

## 4.2 VISUAL-MOTOR CONTROL TASKS

We experimentally evaluate the performance of FORBES on Reinforcement Learning on a variety of visual-motor control tasks from the DeepMind Control Suite (Tassa et al., 2018), illustrated in Figure 6. Across all the tasks, the observations are $64 \times 64 \times 3$ images. These environments provide different challenges. The Cartpole-Swingup task requires a long planning horizon and memorizing the state of the cart when it is out of view; Finger-Spinning includes contact dynamics between the finger and the object; Cheetah-Run exhibits high-dimensional state and action spaces; the Walker-Walk and Walker-Run are challenging because the robot has to learn to first stand up and then walk; Hopper Stand is based on a single-legged robot, which is sensitive to the reaction force on the ground and thus needs more accurate control. As for baselines, we include the scores for A3C Mnih et al. (2016) with state inputs (1e9 steps), D4PG Barth-Maron et al. (2018) (1e9 steps), PlaNet (Hafner et al., 2019b) (1e6 steps) and Dreamer Hafner et al. (2019a) with pixel inputs. All the scores of baselines are aligned with the ones reported in Hafner et al. (2019a) (see details in Appendix A.3). The details of the implementations can be found in Appendix A.2.

Our experiments empirically show that FORBES achieves superior performance and sample efficiency on challenging visual-motor control tasks. As illustrated in Figure 6, FORBES achieves

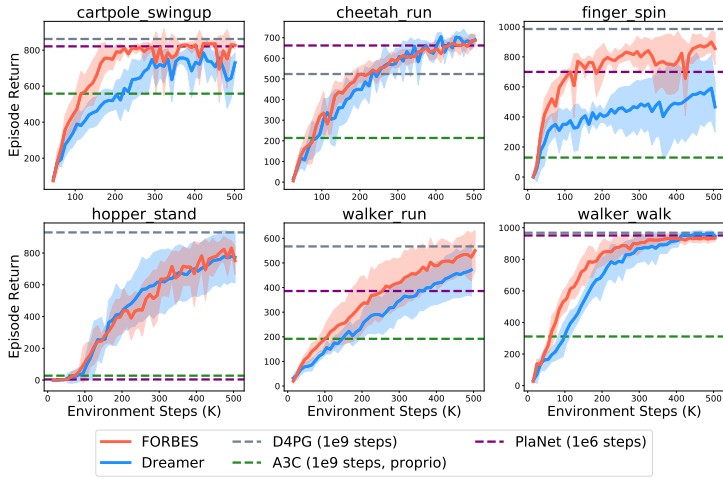

Figure 6: Performance on DeepMind Control Suite. The shaded areas show the standard deviation across 3 seeds. FORBES achieves better performance and sample efficiency in various challenging tasks.

higher scores than Dreamer (Hafner et al., 2019a) in most of the tasks and achieves better performance than PlaNet (Hafner et al., 2019b) within much fewer environment steps. We provide some insights into the results. As shown in Section 4.1, baselines with Gaussian assumptions may suffer from the mixed-up belief and empty belief issues, while FORBES can better capture the general belief states. Furthermore, multiple imagined trajectories can better utilize the diversity in the rollout. Therefore, the inner coherency within the model components allows the agent a better performance. We further discuss the role of multiple imagined trajectories in the next section.

## 4.3 ABLATION STUDY

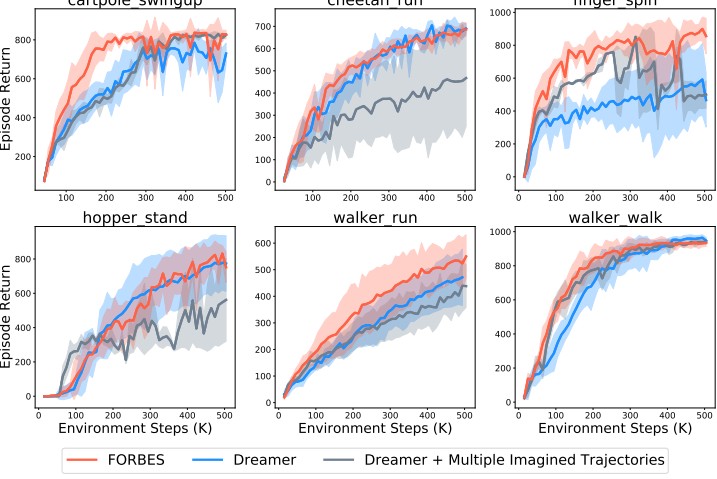

Figure 7: Comparison of the performance between FORBES and Dreamer with multiple imagined trajectories.

In order to verify that the outperformance of FORBES is not simply due to increasing the number of imagined trajectories, we conducted an ablation study in this section. We compare FORBES with the "Dreamer + multiple imagined trajectories" baseline by increasing the number of imagined trajectories in Dreamer to the same as in FORBES. As shown in Figure 7, no consistent and obvious gain can be observed after increasing the number of trajectories to Dreamer. The agent gains slight improvements in two environments and suffers from slight performance loss on other tasks. This result indicates that increasing the number of imagined trajectories may only be effective when the agent can make diverse predictions as in FORBES. The Gaussian assumptions lead to the lack of trajectory diversity, so that increasing the number of imagined trajectories will not effectively help.

## 5 RELATED WORK

**POMDP:** POMDP solving approaches can be divided into two categories based on whether their state, action and observation spaces are discrete or continuous. Discrete space POMDP solvers, in general, either approximate the value function using point-based methods (Kurniawati et al., 2008; Shani et al., 2013) or using Monte-Carlo sampling in the belief space (Silver & Veness, 2010; Kurniawati & Yadav, 2016) to make the POMDP problem tractable. Continuous space POMDP solvers often approximate the belief states as a distribution with few parameters (typically Gaussian) and solve the problem analytically either using gradients (Van Den Berg et al., 2012; Indelman et al., 2015) or using random sampling in the belief space (Agha-Mohammadi et al., 2014; Hollinger & Sukhatme, 2014). However, the classical POMDP methods mentioned above are all based on an accurately known dynamic model, which is a restricted assumption in many real world tasks.

**MBRL for visual-motor control:** Recent researches in model-based reinforcement learning (MBRL) for visual-motor control provides promising methods to solve POMDPs with high-dimensional continuous space and unknown models since visual-motor control tasks can be naturally modelled as POMDP problems. Learning effective latent dynamics models to solve challenging visual-motor control problems is becoming feasible through advances in deep generative modeling and latent variable models (Krishnan et al., 2015; Karl et al., 2016; Doerr et al., 2018; Buesing et al., 2018; Ha & Schmidhuber, 2018; Hafner et al., 2019b;a). Among which, the recurrent state-space model (RSSM) based methods (Hafner et al., 2019b;a) provide a principled way to learn continuous latent belief states for POMDPs by variational inference and learns behaviours based on the belief states using model-based reinforcement learning, which achieves high performance on visual-motor control tasks. However, they assume the belief states obey diagonal Gaussian distributions. Such assumptions impose strong restrictions to belief inference and lead to limitations in practice, including mode collapse, posterior collapse and object vanishing in reconstruction(Bowman et al., 2016; Salimans et al., 2015; Okada & Taniguchi, 2020). A few works propose particle filter based methods that use samples to approximate the belief states (Ma et al., 2020b; Igl et al., 2018). However, they suffer from insufficient sample efficiency and performance.

**Normalizing Flows:** Normalizing Flows (NF) are a family of generative models which produce tractable distributions with analytical density. For a transformation $f : \mathbf{R}^D \to \mathbf{R}^D$, the computational time cost of the log determinant is $\mathcal{O}(D^3)$. Thus most previous works choose to make the computation more tractable. Rezende & Mohamed (2015); van den Berg et al. (2019) propose to use restricted functional form of $f$. Another choice is to force the Jacobian of $f$ to be lower triangular by using an autoregressive model (Kingma et al., 2016; Papamakarios et al., 2018). These models usually excel at density estimation, but the inverse computation can be time-consuming. Dinh et al. (2015; 2017); Kingma & Dhariwal (2018) propose the coupling method to make the Jacobian triangular and ensure the forward and inverse can be computed with a single pass. The applications of NF include image generation (Ho et al., 2019; Kingma & Dhariwal, 2018), video generation (Kumar et al., 2019) and reinforcement learning (Mazoure et al., 2020; Ward et al., 2019; Touati et al., 2020).

## 6 CONCLUSION

General continuous belief states inference is a crucial yet challenging problem in high-dimensional Partially Observable Markov Decision Process (POMDP) problems. In this paper, we propose the **FlO**w-based **R**ecurrent **BE**lief **S**tate model (FORBES) that can learn general continuous belief states by incorporating normalizing flows into the variational inference framework and then effectively utilize the learned belief states in downstream RL tasks. We show that theoretically, our method can accurately learn the true belief states and we verify the effectiveness of our method in terms of both the quality of learned belief states and the final performance of our extended POMDP RL framework on two visual input environments. The digit writing tasks demonstrate that our method can learn general belief states that enable precise and multi-modal predictions and high-quality reconstructions. General belief inference plays a vital role in solving the POMDP, and our method paves a way towards it. In the future, we will explore further approaches to improve the accuracy of belief states inference and information seeking, such as combining contrastive learning and using advanced network architectures such as transformers to build normalizing flows.

## 7  REPRODUCIBILITY

We implement FORBES based on the Pytorch version of Dreamer, provided by "`https://github.com/yusukeurakami/dreamer-pytorch`" and our code will be released after the author notification in "`https://anonymous.4open.science/r/Implementation-of-FORBES-E45B`". We describe the implementation details as well as the hyperparameters in Appendix A.2. We train our FORBES model on NVIDIA Tesla V100 with Pytorch 1.5.0.

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

## A  APPENDIX

### A.1  DETAILS OF AFFINE COUPLING LAYER FOR NORMALIZING FLOW

In this section, we will introduce the details about the affine coupling layer (Dinh et al., 2017).

In the forward function, we split the input $\mathbf{x} \in \mathbb{R}^D$ into two parts according to the dimension: $\mathbf{x} = [\mathbf{x}_{1:k}, \mathbf{x}_{k+1:D}]$. Then, we let the first part $\mathbf{x}_{1:k}$ stay identical, so that the first $k$ dimensions in the output $\mathbf{y} \in \mathbb{R}^D$ is $\mathbf{y}_{1:k} = \mathbf{x}_{1:k}$. After that, we use the identical part as the inputs to determine the transform parameters. In our case, we define two neural network $s, t : \mathbb{R}^k \to \mathbb{R}^{D-k}$, which stand for scale and translation functions. They receive $\mathbf{x}_{1:k}$ as inputs and output the affine parameters. As in (Dinh et al., 2017), the second part can be derived by:

$$\mathbf{y}_{k+1:D} = \mathbf{x}_{k+1:D} \odot \exp(s(\mathbf{x}_{1:k})) + t(\mathbf{x}_{1_k}) \tag{15}$$

Finally, the output, $\mathbf{y}$ is the concatenation of the two parts: $\mathbf{y} = [\mathbf{y}_{1:k}, \mathbf{y}_{k+1:D}]$.

The affine coupling layer is an expressive transformation with easily-computed forward and reverse passes. The Jacobian of affine coupling layer is a triangular matrix, and its log determinant can also be efficiently computed.

### A.2  HYPER PARAMETERS AND IMPLEMENTATION DETAILS

**Network Architecture**  We use the convolutional and deconvolutional networks that are similar to Dreamer(Hafner et al., 2019a), a GRU (Cho et al., 2014) with 200 units in the dynamics model, and implement all other functions as two fully connected layers of size 200 with ReLU activations. Base distributions in latent space are 30-dimensional diagonal Gaussians with predicted mean and standard deviation. As for the parameters network , we use a residual network composed of one fully connected layer, one residual block, and one fully connected layer. The residual network receives $\mathbf{x}_a$ and $c$ as input. The input is first concatenated with the context and passed into the network. The residual block passes the input through two fully connected layers and returns the sum of the input and the output. Finally the last layer outputs the parameters and we use 5 layers of affine coupling flows with a LU layer between them.

In our case, we use samples from the belief distribution as the inputs to the actor and value function as an approximation to the actor and value function with belief distribution as input. Calculating $V(b)$ needs to integrate through both the observation model and state transition model. Our approximation makes an assumption like in Qmdp, to avoid integrating through the observation model.

We use a GRU as the recurrent neural network to summary to temporal information. We assume an initial state $s_0$ to be a zero vector. After taking action $a_t$, we concatenate $a_t$ with the previous state $s_t$ and pass it through a small MLP to get $y_t = f(s_t, a_t)$, and use it as the input to the GRU: $h_{t+1}, z_{t+1} = GRU(h_t, y_t)$. We pass $z_{t+1}$ through an MLP to get the base prior belief distribution $p_0$ (mean and variance) and then we sample from $p_0$ and pass it through a sequence of Normalizing Flow to get a sample from $p_K$. For the posterior distribution, we first use a CNN as encoder to encode the observation $o_t$ into the feature $x_t$, and then concatenate $z_{t+1}$ and $x_t$ and pass them through an MLP to get the base posterior belief distribution $q_0$ and a sequence of Normalizing Flow. Similarly, we finally get a sample $s_{t+1}$ from $q_K$.

**Training Details**   We basically adopt the same data buffer updating strategy as in Dreamer Hafner et al. (2019a). First, we use a small amount of $S$ seed episodes ($S = 5$ in DMC experiments) with random actions to collect data. After that, we train the model for $C$ update steps ($C = 100$ in DMC experiment) and conduct one additional episode to collect data with small Gaussian exploration noise added to the action. Algorithm 1 shows one update step in $C$ update steps. After $C$ update steps, we conduct one additional episode to collect data (this is not shown in Algorithm 1). When the agent interacts with the environment, we record the observations, actions, and rewards of the whole trajectory $((o_t, a_t, r_t)_{t=1}^T)$ and add it to data buffer $\mathcal{B}$.

**Hyperparameters**   For DMControl tasks, we pre-process images by reducing the bit depth to 5 bits and draw batches of 50 sequences of length 50 to train the FORBES model, value model, and action model models using Adam (Kingma & Ba, 2014) with learning rates $\alpha_0 = 5 \times 10^{-4}$, $\alpha_1 = 8 \times 10^{-5}$, $\alpha_2 = 8 \times 10^{-5}$, respectively and scale down gradient norms that exceed 100. We clip the KL regularizers in $\mathcal{J}_{Model}$ below 3.0 free nats as in Dreamer and PlaNet. The imagination horizon is $H = 15$ and the same trajectories are used to update both action and value models. We compute the TD-$\lambda$ targets with $\gamma = 0.99$ and $\lambda = 0.95$. As for multiple imagined trajectories, we choose $N = 4$ across all environments.

For digit writing experiments in Section 4.1, we decrease the GRU hidden size to be 20, let the base distributions be a 2-dimensional diagonal Gaussian and only use 3 layers of affine coupling flows. For the image processing, we simply divide the raw pixels by 255 and subtract 0.5 to make the inputs lie in $[-0.5, 0.5]$.

## A.3   EXTENDED INFORMATION OF BASELINES

For model-free baselines, we compare with D4PG (Barth-Maron et al., 2018), a distributed extension of DDPG, and A3C (Mnih et al., 2016), the distributed actor-critic approach. D4PG is an improved variant of DDPG (Lillicrap et al., 2015) that uses distributed collection, distributional Q-learning, multi-step returns, and prioritized replay. We include the scores for D4PG with pixel inputs and A3C (Mnih et al., 2016) with vector-wise state inputs from DMCcontrol. For model-based baselines, we use PlaNet (Hafner et al., 2019b) and Dreamer (Hafner et al., 2019a), two state-of-the-art model-based RL. PlaNet (Hafner et al., 2019b) selects actions via online planning without an action model and drastically improves over D4PG and A3C in data efficiency. Dreamer (Hafner et al., 2019a) further improve the data efficiency by generating imaginary rollouts in the latent space.

## A.4   AN ABLATION STUDY ON THE NUMBER OF IMAGINED TRAJECTORIES

Figure 8: An ablation study on the effect of different $N$ on DMC environments.

To show the effect of $N$, we adjust the number of imagined trajectories on some DMC environments. We choose $N = 1, 2, 4$ and run 500K environment steps. We run $N = 1, 2$ with 3 different seeds, and $N = 4$ with 5 different seeds (we use the main DMC experiment results, where $N = 4$ here). The result shows that, in Finger Spin, the performance gain caused by multiple imagined trajectories is obvious. In finger spin, there are two objects and their interactions may result in complex loco-motion patterns. When the environmental locomotion pattern itself is complex and flexible enough to incorporate diverse possibilities, then using FORBES allows the agent to make diverse predictions and using the multiple imagined trajectories technique will further exploit the advantages of FORBES. However, not all environments can show the advantages of multiple imaginations. In other environments, where there's only one agent and its behavior is relatively unimodal, a larger $N$ does not effectively improve the performance, and different $N$ results in similar performances.

## A.5 EXTENDED RESULTS ON DMC

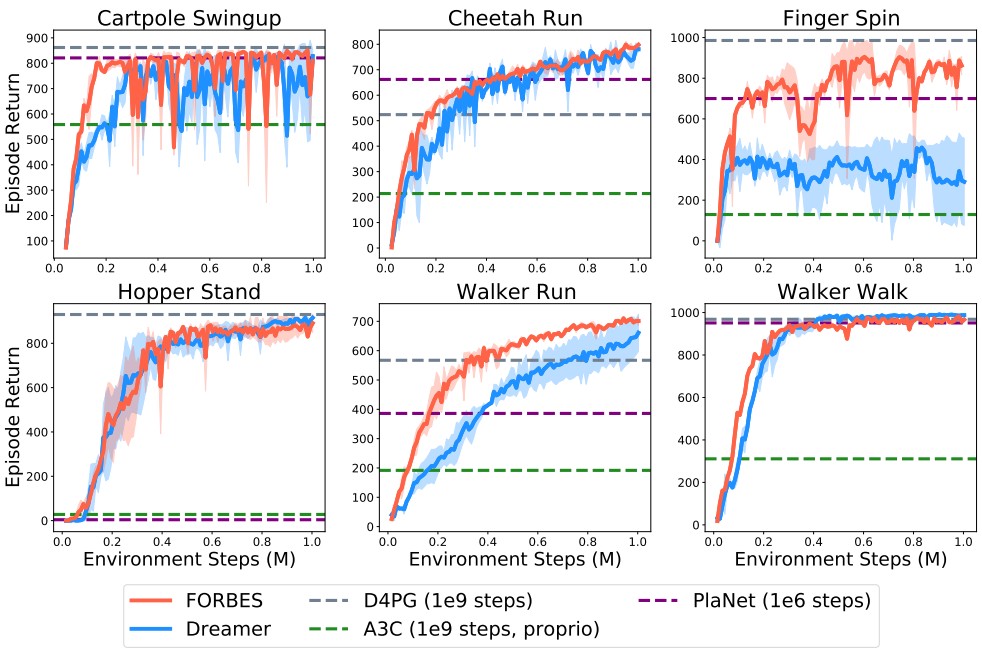

Figure 9: The training curve on DMC environment for 1M environment steps.

We run our algorithm for 1M environment steps and show the curve in Figure 9. We choose 1M environment steps because most of the curves have converged in most of the environments. FORBES achieves higher scores than Dreamer in most of the tasks.

## A.6    AN ABLATION STUDY ON THE MODEL PARAMETERS

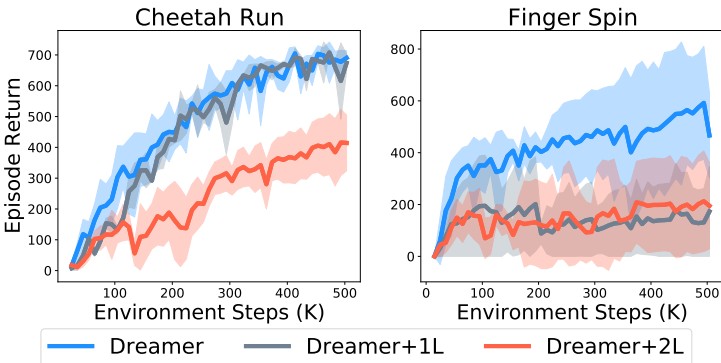

Figure 10: An ablation study on the effect of adding parameters to Dreamer on two DMC environments.

In this section, we show that having a flexible belief state distribution is the key to improving performance, rather than introducing more parameters. Having more parameters do not necessarily mean better performance. Increasing parameters may also make it difficult to converge and negatively affect the sample efficiency.

We add an ablation study that adds more parameters to Dreamer to test the effectiveness of having more parameters. We add $1, 2$ hidden layer(s) to all the MLP in RSSM, and the result is shown in Figure 10. The results show that simply adding parameters cannot improve the performance.

A.7    COMPARISON OF ELBO ON FORBES AND RSSM ON DMC

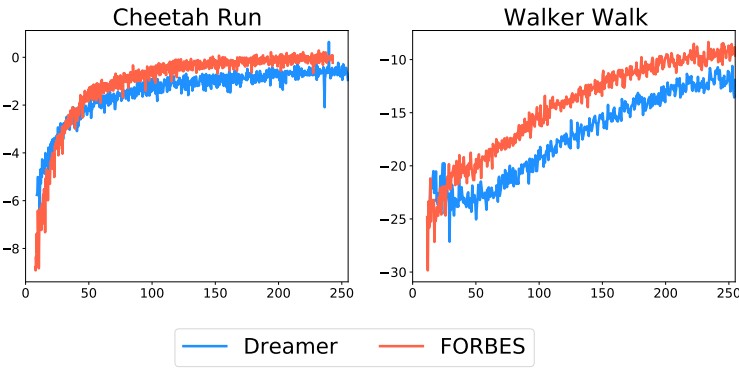

Figure 11: The ELBO of FORBES and RSSM.

We provide the ELBO in DMC environments and FORBES in Figure 11, and FORBES has higher ELBO.

## A.8 EVIDENCE LOWER BOUND DERIVATIONS

The variational bound for latent dynamics models $p\left(o_{1:T}, s_{1:T} \mid a_{1:T}\right) = \prod_t p(s_t|s_{t-1}, a_{t-1})p(o_t|s_t)$ and a variational posterior $q\left(s_{1:T} \mid o_{1:T}, a_{1:T}\right) = \prod_t q\left(s_t \mid o_{\leq t}, a_{<t}\right)$ follows from importance weighting and Jensen's inequality as shown,

$$
\begin{aligned}
\log p\left(o_{1:T} \mid a_{1:T}\right) &= \log \mathrm{E}_{p(s_{1:T}|a_{1:T})}\left[\prod_{t=1}^{T} p\left(o_t \mid s_t\right)\right] \\
&= \log \mathrm{E}_{q(s_{1:T}|o_{1:T},a_{1:T})}\left[\prod_{t=1}^{T} p\left(o_t \mid s_t\right) p\left(s_t \mid s_{t-1}, a_{t-1}\right) / q\left(s_t \mid o_{\leq t}, a_{<t}\right)\right] \\
&\geq \mathrm{E}_{q(s_{1:T}|o_{1:T},a_{1:T})}\left[\sum_{t=1}^{T} \log p\left(o_t \mid s_t\right) + \log p\left(s_t \mid s_{t-1}, a_{t-1}\right) - \log q\left(s_t \mid o_{\leq t}, a_{<t}\right)\right]
\end{aligned}
$$
(16)

A.9 PROOFS OF THEOREM

**Theorem 1:**The approximation error of the lower bound is

$$\log p(o_{1:T}, r_{1:T}|a_{1:T}) - \mathcal{J}_{\text{Model}} = \mathbb{E}_{q_K(s_{1:T}|\tau_T,o_T)}\left[\sum_{t=1}^{T} D_{\text{KL}}(q(s_t|\tau_t,o_t)\|p(s_t \mid \tau_t, o_t))\right]$$

where $p(s_t \mid \tau_t, o_t)$ is the true posterior.

**Proof:**

$$D_{\text{KL}}(q(s_t \mid \tau_t, o_t)\|p(s_t \mid s_{t-1}, a_{t-1}, o_t)) \mid a_{1:T}$$

$$= \int q(s_t \mid \tau_t, o_t) \log \frac{q(s_t \mid \tau_t, o_t)}{p(s_t \mid s_{t-1}, a_{t-1}, o_t)}\text{d}s_t$$

$$= \int q(s_t \mid \tau_t, o_t) \log \frac{q(s_t \mid \tau_t, o_t)}{\frac{p(s_t|s_{t-1},a_{t-1})p(o_t|s_t)}{p(o_t|a_{1:T})}}\text{d}s_t$$

$$= \int q(s_t \mid \tau_t, o_t) \log q(s_t \mid \tau_t, o_t)\text{d}s_t + \int q(s_t \mid \tau_t, o_t) \log p(o_t \mid a_{1:T})\text{d}s_t$$

$$\quad - \int q(s_t \mid \tau_t, o_t) \log[p(s_t \mid s_{t-1}, a_{t-1})p(o_t \mid s_t)]\text{d}s_t$$

$$= \log p(o_t \mid a_{1:T}) + \int q(s_t \mid \tau_t, o_t) \log q(s_t \mid \tau_t, o_t)\text{d}s_t - \int q(s_t \mid \tau_t, o_t) \log[p(s_t \mid s_{t-1}, a_{t-1})p(o_t \mid s_t)]\text{d}s_t$$

$$= \log p(o_t \mid a_{1:T}) + \int q(s_t \mid \tau_t, o_t) \log q(s_t \mid \tau_t, o_t)\text{d}s_t - \int q(s_t \mid \tau_t, o_t) \log p(s_t \mid s_{t-1}, a_{t-1})\text{d}s_t$$

$$\quad - \int q(s_t \mid \tau_t, o_t) \log p(o_t \mid s_t)\text{d}s_t$$

$$= \log p(o_t \mid a_{1:T}) + \int q(s_t \mid \tau_t, o_t) \log \frac{q(s_t \mid \tau_t, o_t)}{p(s_t \mid s_{t-1}, a_{t-1})}\text{d}s_t - \int q(s_t \mid \tau_t, o_t) \log p(o_t \mid s_t)\text{d}s_t$$

$$= \log p(o_t \mid a_{1:T}) + D_{\text{KL}}\left(q\left(s_t \mid \tau_t, o_t\right)\|p\left(s_t \mid s_{t-1}, a_{t-1}, o_t\right)\right) - \mathbb{E}_{q(s_{1:t}|\tau_t,o_t)}[\log p\left(o_t \mid s_t\right)]$$

$$\tag{17}$$

For a sequence from time 1 to T, we have

$$\sum_t D_{\text{KL}}\left(q\left(s_t \mid \tau_t, o_t\right)\|p\left(s_t \mid s_{t-1}, a_{t-1}, o_t\right)\right)$$

$$= \log p(o_{1:T} \mid a_{1:T}) - \mathbb{E}_{q(s_{1:t}|\tau_t,o_t)}\left[\sum_{t=1}^{T}(\log p(o_t|s_t) - D_{\text{KL}}(q(s_t|\tau_t,o_t)\|p(s_t|s_{t-1},a_{t-1})))\right]$$

$$\tag{18}$$

Then we can derive the Theorem 1 with equation 18:

$$\log p(o_{1:T}, r_{1:T} \mid a_{1:T})$$

$$= \mathbb{E}_{q_K(s_{1:T}|\tau_T,o_T)}\left[\sum_t D_{\text{KL}}\left(q\left(s_t \mid \tau_t, o_t\right)\|p\left(s_t \mid s_{t-1}, a_{t-1}, o_t\right)\right)\right]$$

$$\quad + \mathbb{E}_{q(s_{1:T}|\tau_T,o_T)}\left[\sum_{t=1}^{T}(\log p(o_t|s_t) + \log p(r_t|s_t) - D_{\text{KL}}(q(s_t|\tau_t,o_t)\|p(s_t|s_{t-1},a_{t-1})))\right]$$

$$= \mathbb{E}_{q_K(s_{1:T}|\tau_T,o_T)}\left[\sum_t D_{\text{KL}}\left(q\left(s_t \mid \tau_t, o_t\right)\|p\left(s_t \mid s_{t-1}, a_{t-1}, o_t\right)\right)\right] + \mathcal{J}_{\text{Model}}$$

$$= \mathbb{E}_{q_K(s_{1:T}|\tau_T,o_T)}\left[\sum_t D_{\text{KL}}\left(q\left(s_t \mid \tau_t, o_t\right)\|p\left(s_t \mid \tau_t, o_t\right)\right)\right] + \mathcal{J}_{\text{Model}}$$

$$\tag{19}$$

where $p(s_t \mid s_{t-1}, a_{t-1}, o_t) = p(s_t \mid \tau_t, o_t)$ given the sampled $s_{t-1}$ from $q(s_{1:t}|\tau_t, o_t)$.

## A.10 MORE RESULTS ON DIGIT WRITING EXPERIMENTS

In this section, we show more results of the predictions on the digit writing experiment in Figure 12.

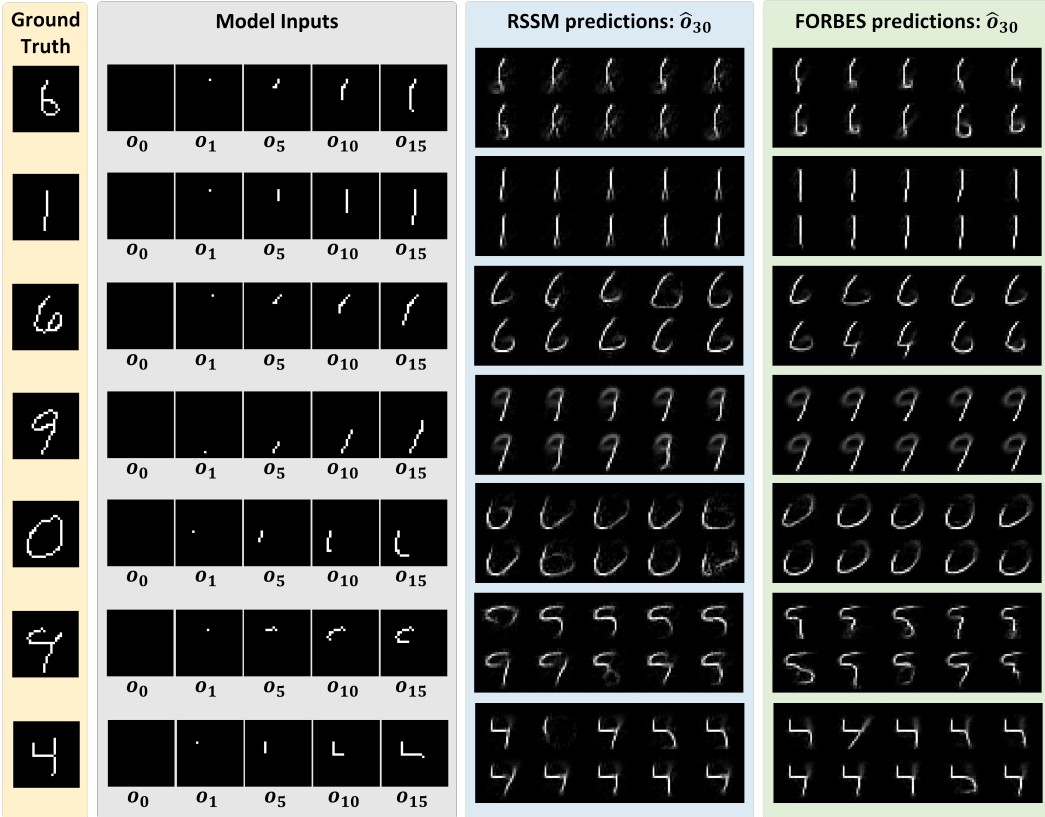

Figure 12: Additional prediction results on sequential MNIST of two models.

## A.11 Reconstructions of the visual control tasks

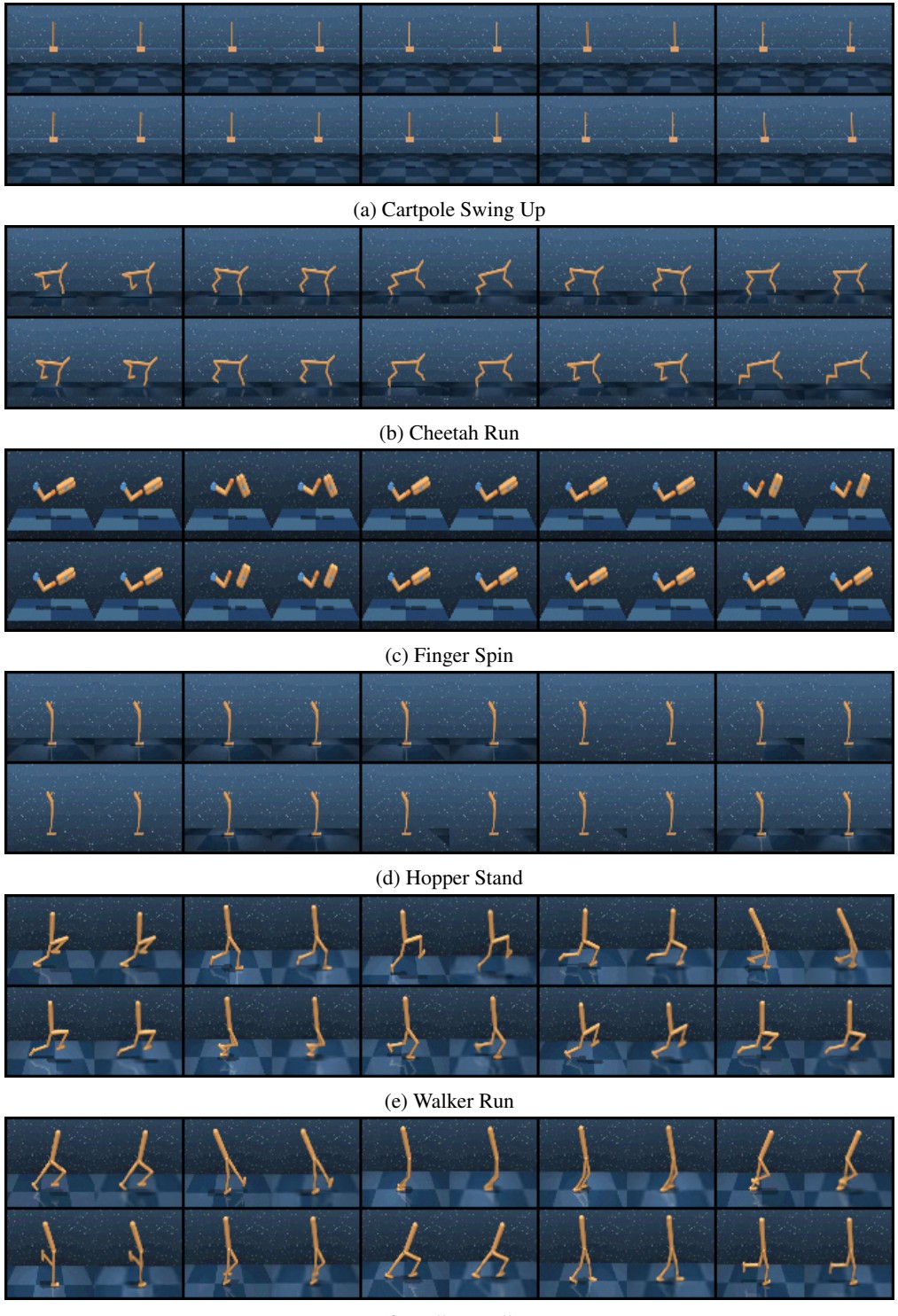

(a) Cartpole Swing Up

(b) Cheetah Run

(c) Finger Spin

(d) Hopper Stand

(e) Walker Run

(f) Walker Walk

Figure 13: The reconstruction results on of FORBES six environments from DeepMind Control Suite(Tassa et al., 2018).

In this section, we show the reconstructions of the visual control tasks during the evaluating phase.

For each environment, we use 10 frames. The left one is the original picture for each frame, and the right one is the reconstruction picture. The following results in Figure 13 show that FORBES can make high-quality reconstructions. The corresponding videos can be found in the supplementary material.

