# OpenReview forum: "Flow-based Recurrent Belief State Learning for POMDPs"
_ICLR.cc/2022/Conference — ICLR 2022 Submitted_

### Official Review · Reviewer_9het · 2021-10-26

**Correctness:** 3
**Technical Novelty And Significance:** 3
**Empirical Novelty And Significance:** 2
**Recommendation:** 8
**Confidence:** 2

**Main Review:**

Positives:
- The paper demonstrates that normalizing flows can improve control performance and sample efficiency in MBRL over approaches working with a Gaussian assumption.
- While VAEs and other generative models have been applied to belief inference, this appears to be the first work applying NFs. It is not groundbreaking novelty, but seems like a solid evaluation of the idea.
- One of the advantages of POMDP planning is the ability to reason about future uncertainty. The sequence MNIST result where the proposed method is able to predict either 3's or 7's is quite exciting, since it shows the method has some capability for such reasoning. This aspect would be interesting to see even more details about.

Negatives:
- There are several missing baselines and comparisons in the empirical results, and some lack of clarity about parameter choices (details below).
- Throughout, the paper seems to conflate in writing and symbol use the belief state and the hidden state. This is especially prevalent in Sect. 3.2., but should be revised overall. For conceptual clarity, the paper should in my opinion be careful to separate 1) hidden states or samples of hidden states (typically denoted s in POMDP literature) 2) belief states, or distributions over the hidden state, and their approximations (typically denoted b in POMDP literature).


Detailed comments:


There are a couple of omissions in the evaluation, which should be justified or a comparison added:
- The paper does not compare to PlaNet-Bayes (Okada et al., 2020) in the control tasks. This is the previous state-of-the-art in the area. Since the paper does compare to PlaNet-Bayes on the MNIST setting, I am wondering if it would not be possible and reasonable to compare on the control tasks as well.
- Igl et al. (2018) "Deep Variational Reinforcement Learning for POMDP" is also very closely related to the proposed method. A comparison here would also be welcome, or some arguments why a comparison is not needed.
- In some works addressing partially observable RL, belief states are inferred by using a recurrent neural network directly on the action-observation history. It would be nice to see a simple RNN baseline in the control tasks.
- A missing ablation is how the number of imagined trajectories N affects the results. This should ideally be done in a domain where Gaussian or other simplifying assumptions on belief inference do not work, otherwise even with a single trajectory results could be OK. I also could not find information on how many trajectories were used in the experiments.
- As mentioned in the strengths, it would be great to have some further details on the method's capability for reasoning about future uncertainty. For example, is the proposed method better than RSSM always, or especially for long prediction horizons?


Other comments & minor:
- Theorem 2 seems to not seem to add much to the paper, and does not merit a full reproduction. A citation could be sufficient.
- Figure 3 needs some improvements. It is not really clear what the colors here are indicating, or what happens at the intersections of the arrows.
- In 2.1., it would be helpful to define the POMDP objective considered. Is it maximization of expected sum of discounted rewards?
- "Dream to control" is duplicated in bibliography

**Post-rebuttal comments:**
I thank the authors for answering my questions and concerns regarding the paper. The author rebuttal does a good job addressing my concerns on presentation and ablations. I am raising my score to reflect this. However, I also agree with many of the concerns raised by other reviewers regarding the significance of the work.



**Summary Of The Paper:**

This work addresses model-based reinforcement learning (MBRL) in partially observable visual control tasks. The principal novelty of the work is the use of normalizing flows (NFs) in the belief inference model. The paper derives the evidence lower bound (ELBO) for the proposed inference model, and presents a related RL framework. Empirically, the proposed model's inference capabilities are tested in the sequential MNIST domain, and its control performance is evaluated in six continuous control problems. The results indicate improved belief fidelity, and improved sample complexity for learning to solve the control tasks.

**Summary Of The Review:**

This paper proposes to use normalizing flows as belief representations in a POMDP, addressing drawbacks of VAEs or diagonal Gaussian approximations in representing multi-modal beliefs. While not highly original, the work is technically sound, and provides the necessary empirical evidence to merit acceptance. Nevertheless, I encourage the authors to address the concerns raised by other reviewers as well, and to provide comparisons to related works the authors also agree are highly relevant.

---

### Official Review · Reviewer_8rin · 2021-10-31

**Correctness:** 4
**Technical Novelty And Significance:** 2
**Empirical Novelty And Significance:** 2
**Recommendation:** 6
**Confidence:** 4

**Main Review:**

* Strengths
The main contribution of this work is very straightforward: using a richer parametric distribution to learn environment dynamics leads to better sampled trajectories, and an algorithm that can optimize the policy of an agent more efficiently. I think the simplicity of the approach is the strength of the algorithm, and the method is tested and compared on a well known control tasks suite. Results show a consistent improvement in sample effiency of the RL framework proposed over closely related pieces of work and other standard RL algorithms (e.g. D4PG and A3C).

* Weaknesses
1) While most of this work is well written and the method is clearly described, I don't think the proposed contributions is particularly novel. In particular, the idea of using more flexible distributions for POMDPs have been explored (as mentioned by the authors) in the form of particle filters. While these methods are referenced (e.g. Ma et. al. 2020b, Igl et al. 2018), they are not compared or seriously discussed beyond an unsubstantiated claim that they suffer from insufficient sample efficiency. The authors also fail to cite an even more closely related work that directly addresses (and in my eyes, go beyond what the authors explore in this paper), see Gregor et al. Gregor et al. not only also suggests using normalizing flows, but find that using a convolutional DRAW (VAE model) outperforms flows for learning a model of complex environment dynamics.

2) The second contribution proposed, namely the POMDP RL Framework is an effective way to leverage a model of the environment, as it allows to use imaginary trajectories to train TD(lambda) actor and critics. However, I fail to see exactly how this is different from the referenced Dreamer (Hafner et al. 2019a) work. The authors should make it clear how it differs from Dreamer beyond the use of normalizing flows. If there are no differences, then it should be clearly stated that they make use of an existing method. If there are differences, then I'd like to see further experiments to understand how these differences affect the algorithms performance.

3) The FORBEs algorithm lacks detail on how the data buffer B is created, i.e. how is data collected during training? See PlaNet and Dreamer works for examples of algorithm description wrt. how experience buffers are updated as the agents train. Also, how are tau_t (which includes a_<t and o_<t) obtained if they are apparently not in the batch B?

4) What number N of sampled trajectories are used during training? Furthermore, it would be interesting to see the performance of the method for different number N, to get a sense of how useful it is to increase the number of trajectories and whether there are diminishing returns.

Some other minor issues:
- 3 seeds seems like a small number and some experiments still have large error bars (e.g. Cheetah Run and Walker Run). I suggest increasing the number of runs from 5 to 10 if possible.
- When comparing with PlaNet, it would be more informative to also show PlaNets training curve instead of converged performance after an arbitrary number of steps (1e6 steps). It may well be that PlaNet is just as sample efficient but somehow plateaued/converged a bit slower towards the end of training.

**Post-rebuttal update**

I appreciate the willingness of the authors to discuss and address our concerns, and their efforts to improve their work. The paper is now clearer in its contributions, limitations and additional experimental results and details will make it more useful to the community.
I've revised the score to reflect all these improvements, but I still think this work missed out on an opportunity to truly tackle the challenges of POMDP: handling environment stochasticity and state uncertainty. During our discussion, the authors have agreed that the method makes some assumptions that limit learning policies and values that can reason about state uncertainty (which is now better captured thanks to the normalizing flow) -- it seems that this would be the main reason for using the proposed flow-based model.
Instead, the environments in the experiments have essentially no stochastic temporal dynamics. I feel the updated paper still largely sidesteps discussing the above limitation.
Another minor point, is that I find it a bit puzzling that Dreamer+multiple imagined trajectories seemingly fairs quite a bit worse than Dreamer (Fig. 7) -- this is a bit counterintuitive (I would have thought that at worst, performance would be the same) and I would've liked further explanation.

* References:

Gregor et al. Shaping Belief States with Generative Environment Models for RL. NeurIPS 2019

**Summary Of The Paper:**

This paper proposes learning belief states for POMDPs using flexible posteriors and prior distributions for the state-space model. The authors state that related work often restrict the distributions to conditional diagonal Gaussians which causes the model to underfit the data. They propose a flow-based model to solve the well known issues of using inflexible distributions (i.e. inability of the aggregate posterior to match the prior) which can cause the model to generate implausible samples. In the context of POMDPs, the authors propose using the learned model of environment dynamics to train an actor and critic by sampling trajectories, resulting in a highly sample efficient RL algorithm. Results show the benefits of using the normalizing flow compared to baselines that only use a Gaussian distribution.

**Summary Of The Review:**

It is well known that VAEs with diagonal Gaussian distributions for the prior and posterior do not fit many datasets well. There are a large number of deep generative models proposed in the literature that are much more flexible and suited to different domains. The authors propose using one of these, normalizing flows, and apply it to learning a model of the environment dynamics in POMDPs settings, and in turn use it to train agents in a sample efficient way. While I think this is a good idea, and experiments indeed show the benefits of using such a model, the contribution in itself is not very novel, as prior work has suggested using richer models (see referenced works on particile filters, or the work of Gregor et al. which also refers to the idea of using flows). I think this paper would be more interesting if it compared a wide range of flexible models, and included a more extensive set of datasets. Alternatively, the contributions would also be strengthened if it provided novel ways to use the learned models when training agents in POMDP tasks. As it stands, I do not recommend accepting the paper due to its limited novelty.

---

### Official Review · Reviewer_CAct · 2021-11-02

**Correctness:** 4
**Technical Novelty And Significance:** 3
**Empirical Novelty And Significance:** 3
**Recommendation:** 8
**Confidence:** 4

**Main Review:**

The authors propose and successfully use a more powerful form of distributions for learning belief states in POMDPs. To the best of my knowledge this is the first study on normalising flows in this context (the authors do cite other work using normalising flows in RL which do not apply in this context). The sequential MNIST results are convincing, but the results on DM Control Suite are not so strong - having other domains where the benefit of normalising flows are more pronounced would make this more attractive than simply sticking to simpler isotropic Gaussians. FORBES also introduces more learned parameters, but I doubt that this would contribute towards a change in performance.

The major weakness is the significance/novelty of this paper. Is it fair to say that this is Dreamer + normalising flows latent space + multiple imagined trajectories? Knowing the literature and from the experiments it would seem so, but the paper is written in a way that makes it unclear how exactly FORBES differs from prior work. On a side note, it is interesting that Deamer + multiple imagined trajectories does not seem to work that well - possibly due to the unimodal latent space. Further to this, while the DreamerV2 paper is cited with its argument for multimodality, DreamerV2 and its discrete latent space (which is also much more flexible than an isotropic Gaussian) are not discussed - which would seem an omission given the supposed benefit of normalising flows over isotropic Gaussians.

On a minor note, there are several spelling mistakes throughout, e.g., "beilef", "poorfs".

**Post-rebuttal update:** I agree with the authors that FORBES is more general than an alteration to Dreamer, and there are theoretical contributions that are of relevance to the community. However, given my awarereness of the literature and the concerns of the other reviewers, I would hesitate to move my recommendation much higher.

**Post-rebuttal update 2:** The authors have put considerable work into addressing the concerns of the other reviewers, and it is now a more solid paper. Although I would appreciate a more thorough discussion of related work (perhaps in supplementary material), e.g. of why certain baselines were not included, I think it is worth accepting given the updates.

**Summary Of The Paper:**

Belief states are a common solution to learning in POMDPs, and in recent years people have applied deep recurrent generative models to approximating belief states in complex POMDPs. In this paper, the authors go beyond the isotropic Gaussian assumption used in most deep learning works and use normalising flows to represent more flexible and multimodal distributions. The authors demonstrate that their approach is better at modelling multimodal belief states and matches/outperforms Dreamer, which is a natural baseline for their method.

**Summary Of The Review:**

The authors presented a relatively simple improvement to solving POMDPs by replacing isotropic Gaussians with normalising flows. While this does indeed work, the technical and empirical contributions are reasonably significant, and so I would recommend this for an accept.

---

### Official Review · Reviewer_DDeH · 2021-11-06

**Correctness:** 3
**Technical Novelty And Significance:** 3
**Empirical Novelty And Significance:** 2
**Recommendation:** 6
**Confidence:** 4

**Main Review:**

Accurate belief tracking is important but often hard for continous-state POMDP.  This paper proposes a normalizing flow based belief approximation, and learning the belief approximation by maximizing an ELBO. An actor and a critic can be jointly learned together with the normalizing flow belief approximation. The experiments show that the normalizing flow based belief achieves higher ELBO than a baseline RSSM on the MNIST sequence dataset, and the actor-critic algorithm achieves good performance on several problems from DeepMind Control Suite.

The paper is generally easy to follow, and the idea of improving belief tracking by exploiting the power of normalizing flows to represente complex distribution is interesting.

However, some important aspects of the idea is not clear to me, making the correctness of the idea unclear. In particular, comments and clarifications on the following are helpful:
- In Eq. (7) and Theorem 1, expectation should be taken wrt $q(s_{1:T} | \tau_{T}, o_{T})$. Also state clearly how $q(s_{1:T} | \tau_{T}, o_{T})$ factorizes.
- What's the motivation behind the architecture in Fig. 3?
- How exactly are the observations and actions encoded into $q_{\psi}(s_{t} | \tau_{t}, o_{t})$?
- Where do the transition dynamics $p_{\psi}(s_{t} | s_{t-1}, a_{t-1})$ and the observation model $p_{\psi}(o_{t} | s_{t})$ appear in the belief model? If they are used for encoding the actions and observations, then these are not designed to approximate the true transition dynamics and the true observation model, and using them as transition dynamics and observation model in Eq. (12) and Eq. (13) seem unjustified.
- Eq. (9) actually deviates from Eq. (7), because while Eq. (7) uses a Markov transition model $p(s_{t} | s_{t-1}, a_{t-1})$, Eq. (9) uses a non-Markov transition model $p(s_{t} | \tau_{t})$.
- In the RL algorithm, the actor and critic are both functions of the state, not the belief. How is an action for a belief calculated then?
- While there is a description of the learning objectives for the actor and the critic, the overall RL algorithm is not described. For example, how is training experience collected? Which experiences are stored in the experience buffer in Alg. 1? How often are the belief model, the actor and the critic updated?

I also have some questions and concerns over the experiments
- For the digit writing task, what's the state? The details of the networks used seem to be missing too.
- RSSM is from the PlaNet paper, not from the paper of Okada et al. 2020.
- The proposed belief model and RSSM are compared only on the digit writing task. Can they be compared on problems from DeepMind Control Suite?
- For DeepMind Control Suite, only results for 6 out of the 20 problems are reported. For some problems, the results for Dreamer do not yet match those reported in the Dreamer paper yet. This may be because the algorithms are only run for 500k steps, while the results in the Dreamer paper are obtained for 5 million steps. It would be helpful to run the new algorithm longer to see whether it can eventually match Dreamer's performance as reported in the orginal paper.


Minor comments:
- pp. 3: usally -> usually
- In the description of the POMDP model, $s_{1}$ denotes the initial state, and it doesn't produce an observation $o_{1}$, but $o_{1}$ appears in multiple notations.

**Post-rebuttal**

I appreciate the authors' effort in addressing the questions and concerns, and I have increased my score in view of the the improved clarity and more thorough experiments.

A main weakness of the work is that it mainly replaces the latent model in Dreamer with normalizing flows, as noted by other reviewers too. In view of this, the paper could be improved in several aspects.
* Current version is somewhat unclear and misleading about its novelty, and should provide a proper discussion on what it adds on top of PlaNet and Dreamer.
* In addition, since the main novelty is in the use of a different latent model, I find the experiments not as thorough as it should be - it should compare with Dreamer on the same tasks, but results are only presented on a subset of them.
* The authors have addressed most questions on clarification of the work well as far as I can see, but there are a few remaining concerns (e.g., those in other reviewers' post-rebuttal updates), and these need to be discussed/addressed.



**Summary Of The Paper:**

This paper proposes using normalizing flow based belief approximation for continous-state POMDPs and learning the belief approximation via variational inference. In addition, the paper proposes learning an actor-critic reinformcement learning algorithm based on the proposed belief representation.


**Summary Of The Review:**

Interesting idea, but there are some concerns over the motivation, explanation, and experiments.

---

### Decision · Program_Chairs · 2022-01-20

**Decision:**

Reject

**Comment:**

Summary: this is a difficult paper to meta-review, since it contains some insightful ideas and interesting experiments, while it also unfortunately contains omissions, confusions, and places where clarity is lacking (see below). One consistent theme is that the paper is too dismissive of prior work; the exposition is not as clear as it should be about what aspects of FORBES are present in previous papers, it uses too broad a brush to describe prior methods (resulting in too-general statements about what these methods can't do), and it skips important chunks of the extensive literature on POMDP belief representation and tracking. As a result, the paper doesn’t do a good job concisely and accurately stating its contribution; there is still reasonable concern about how significant this contribution is. On the other hand, the experimental results for FORBES are interesting; the new method seems to represent a better combination of techniques than at least many existing works, at least to the resolution of the experiments’ statistical power. So the end question is whether interesting experimental results and a new combination of techniques are enough to outweigh the problems outlined above. In the end we believe that the correct outcome is rejection; but we have every expectation that a future version of the paper will resolve the difficulties outlined here and will appear in a future conference.

A brief note about the discussion: the original scores for this paper were lower. While some reviewers raised their score later in the discussion, a thorough reading of the discussion and the revised paper indicates that a substantial fraction of the issues leading to the lower scores still remain.

More details:

There is a lot of prior work on tracking belief states, which should be cited more thoroughly. The paper's intro makes it sound like diagonal Gaussians were the only previous alternative. At least, the intro should cite older work on MCMC methods like particle filters (e.g., Thrun’s book Probabilistic Robotics, or Arnaud Doucet’s work), and prior deep-net papers that attempt non-Gaussian representations, even if these don’t perform as well as hoped (see below for examples). It is also important to compare to RKHS representations of beliefs, such as Nishiyama, Boularias, Gretton, Fukumizu 2012; these handle multimodality, and can behave similarly to deep nets if they use the neural tangent kernel. Accurately comparing to prior work is one of the most important functions of a paper, so it doesn’t make sense to be unfairly critical of prior work or to skip it.

The paper is also unclear about the effects of Gaussian distributions at different places in a variational approximation. Because of this lack of clarity, the criticisms it levels at previous variational methods seem to be true only of some of them.

In particular, the introduction should distinguish between two uses of Gaussian approximations: first for the belief itself, and second for the distribution of observations given a belief. Some prior works make only one of these approximations. For example, a non-Gaussian distribution used as a belief state can predict multi-modal future behaviors, even if we approximate observations under a given belief as Gaussian.

The introduction should also distinguish between two common places that a Gaussian could enter into a variational approximation: at the input or at the output of a network. A Gaussian latent at the input of a variational network (even if it has diagonal covariance) can result in a highly non-Gaussian output distribution, while Gaussian noise added at the end will (if it is the only noise) lead to a Gaussian output. Again, some of the statements in the intro apply only to the latter use of a Gaussian, while some prior work focuses on the former use.

There is an important conceptual confusion in the paper about what it means to have a multimodal belief state: the paper presents the true belief as an inherent property of an environment, while in fact it is a property of an environment *model*. So, there can be two different equally accurate models of the same environment which differ in the belief representation; a simple example would be to use either a continuous state whose components are joint angles, or a discrete state obtained by finely discretizing this continuous one. In the first case the belief would be a distribution over the continuous space, while in the second it would be a categorical distribution (a point in a simplex). A consequence of such a difference is that beliefs can be multimodal in one representation and not another.

The importance of this confusion is that, since we are asking our network to learn a belief state, the learning process could potentially favor representations that lead to unimodal beliefs — so it’s not clear theoretically that forcing a unimodal belief representation is necessarily a disadvantage. The paper presents the situation as if the disadvantage is forced by theory, while instead the argument should be based on experiments: e.g., one could try to show that unimodal representations, even if given a higher latent dimension to work with, aren’t empirically able to capture the same information.

Some interesting prior deep-net POMDP papers that might need better discussion:
* Han, Doya, and Tani ICLR 2020 (which isn’t cited here) puts the Gaussian latent variable as an input to the network for predicting beliefs (eq 2), resulting in a possibly highly non-Gaussian output representing the belief.
* Tschiatschek et al, 2018 (also not cited) uses a Gaussian *mixture* as the variational distribution to approximate beliefs, again allowing multimodality.
* Igl et al. 2018 (which is cited only late in the current paper, and basically dismissed) uses a deep version of particle filters to allow non-Gaussian distributions for both beliefs and observations.
* More work that is potentially relevant but not adequately compared (even if briefly cited): Gregor et al. (2019), DreamerV2, Ha & Schmidhuber’s World Models. Each of these makes at least some choices to try to handle at least some kinds of multimodality, so a clear explanation of differences that avoids the confusions mentioned above would be very helpful.
* In general, the results of the search “variational encoder POMDP” seem to include a number of papers not cited in the current paper; another useful search is “normalizing flow POMDP"

Finally, in the experiments section, the paper needs to correctly report the reliability of its conclusions. In some places (e.g., Fig. 5) there’s no mention of reliability or repeatability of conclusions; the paper just says that its evidence “support[s] the claim that FORBES can better capture the complex belief states”. In other places (e.g., Fig. 6, 7), the paper displays uncertainty representations based on only a few replications of an experiment (e.g., 3 seeds for Fig. 6, or 5 seeds when a reviewer requested extra experiments). The corresponding uncertainty estimates almost certainly are strongly biased too low (too certain); e.g., three runs would have less than a 50% chance of even seeing failure modes that happen with probability as high as 20% (0.8^3 = 0.512 > 0.5), meaning that the estimated standard deviation could be almost arbitrarily badly biased downward. To be clear, experiments with few replicates can still be highly useful and informative, and it’s true that some experiments are too expensive to run many times; but in such cases the paper should add appropriate caveats to its conclusions. For example, instead of reporting the sample standard deviation based on a normal model, the paper could report a confidence interval based on a more robust model or test, such as a Wilcoxon test. (To illustrate the difference, confidence intervals at typical significant levels like p=0.05 would be vacuous (infinitely wide) under Wilcoxon with 3 seeds, but much-weaker p-values would still yield non-vacuous intervals.)

A few smaller questions:

The authors added a nice ablation study to compare to Dreamer; this is great to see. It would be good to discuss the connection to earlier methods such as PlaNet and Dreamer at places where the current method is similar or different (e.g., different from Dreamer in the belief state representation in sec 2.2, but similar in the RL framework in section 3.2). These comparisons would aid in the reader’s understanding of what is new in FORBES.

An unusual feature of FORBES is that the variational approximation to the belief at time t+1 is not a function of the belief at time t. Instead the belief inference network q_{\psi,\theta} takes as input the entire past trajectory, uses convolution and recurrence to reduce the variable-length input to fixed dimension, and passes this fixed-dimension representation through a normalizing flow mapping. It would be interesting to discuss the reason for this design decision. In particular, it seems like it would inhibit tracking — i.e., it could be hard to propagate information from one belief distribution to the immediate next one, particularly if there are a few unlikely observations scattered through a trajectory.

A minor point for clarity: in Fig 1 it's unclear what distributions the white and gray triangles refer to. They don't seem to correspond to a natural belief state: instead maybe they incorporate three simultaneous observations from the same starting belief? Correctly intersecting beliefs is an important issue though, so at a high level the point that the figure is trying to make fits well.

Another point for clarity: “there always exists a diffeomorphism that can turn one well-behaved distribution into another”: this is true for some definition of ”well-behaved”, but it’s misleading to say it this way. E.g., it is not true if the distributions in question can have atoms, or differ in dimension or topology; these exceptions are unfortunately important cases that do come up in practice.